# Phenotypic Characterization of Quinoa (*Chenopodium quinoa* Willd.) for the Selection of Promising Materials for Breeding Programs

**DOI:** 10.3390/plants10071339

**Published:** 2021-06-30

**Authors:** Elsa Helena Manjarres-Hernández, Diana Marcela Arias-Moreno, Ana Cruz Morillo-Coronado, Zaida Zarely Ojeda-Pérez, Agobardo Cárdenas-Chaparro

**Affiliations:** 1Grupo CIDE Competitividad Innovación y Desarrollo empresarial, Universidad Pedagógica y Tecnológica de Colombia, Tunja 150003, Colombia; ana.morillo@uptc.edu.co; 2Grupo de Investigación BIOPLASMA, Universidad Pedagógica y Tecnológica de Colombia, Tunja 150003, Colombia; diana.arias04@uptc.edu.co (D.M.A.-M.); zaida.ojeda@uptc.edu.co (Z.Z.O.-P.); 3Grupo de Química—Física Molecular y Modelamiento Computacional QUIMOL, Universidad Pedagógica y Tecnológica de Colombia, Tunja 150003, Colombia; acardenasc@unal.edu.co

**Keywords:** grain quality, ancestral crop, *Chenopodium quinoa*, morphologic descriptors, selection index, yield, pseudocereal

## Abstract

Quinoa is an ancestral crop in the Andean region, characterized by its adaptability to different agroclimatic conditions, great nutritional value, and broad genetic variability. A preliminary approach for understanding the genetics of quinoa materials entails a morphologic characterization, which can provide the basis for the selection of materials that satisfy the needs of farmers and consumers. Therefore, this study aimed to evaluate the phenotypic characteristics of thirty genetic *C. quinoa* accessions for the selection of outstanding accessions in terms of yield and grain quality. A randomized complete block design was used, with nine replications for each accession under greenhouse conditions. Nine quantitative and twelve qualitative descriptors were evaluated with descriptive analysis, Spearman correlation variance, and multivariate and cluster analysis. The results showed that the accessions with heights greater than the average (>176.72 cm) and long panicles (>57.94 cm) presented lower yields and smaller seed sizes, thus decreasing the grain quality. The multivariate and cluster analyses established groups of accessions with good yields (>62.02 g of seeds per plant) and stable morphological characteristics. The proposed selection index, based on yield components and morphological descriptors, indicated four accessions as potential parents for quinoa breeding programs in Colombia.

## 1. Introduction

Quinoa (*Chenopodium quinoa* Willd.) is a pseudocereal that is considered one of the most complete foods for humans. It is grown in South America, from Colombia to southern Chile. However, the greatest diversity is found in Peru and Bolivia [1,2]. This species presents high phenotypic variability that can be easily recognized by the pigmentation of the plant, inflorescences and seeds, earliness diversity, shape and size of grain, compaction of the panicles, and resistance to adverse factors, such as drought, frost, excess humidity, salinity, and diseases among others. This variability explains the ability of this species to adapt to different agroclimatic conditions [3,4,5].

According to the Food and Agriculture Organization of the United Nations (FAO) [6], in recent years (2000–2019), there has been a significant global increase in the area cultivated with quinoa crops, mainly in Peru and Bolivia, with increases between 36% and 72%, respectively [4,7]. In Colombia, quinoa production has increased by more than 1100 tons since 2017 [8]. The departments where quinoa is grown include Cundinamarca, Cauca, Nariño, and Boyacá. The latter is in the central, eastern part of Colombia and has a collection of quinoa germplasm [9]. However, information on the morphological characteristics and grain yield of these accessions is lacking.

Therefore, morphoagronomic characterizations using quantitative and qualitative descriptors of domestic and introduced quinoa materials in Colombia are essential to efficiently using genetic variability to increase the productivity of crops under different environmental conditions for the commercialization of grains or derived products in response to global demand for this pseudocereal.

In Colombia, studies on the morphoagronomic characterization of quinoa are scarce. Torres and collaborators carried out a morphological evaluation of 19 accessions on the Bogotá Savanna, finding high variability in terms of grain yield, biomass, and earliness [10]. Veloza and collaborators found differences in the materials Piartal, Nariño and Bolivia in terms of yield, protein content, stalk coloration, panicle shape and physiological maturity [11]. In the Department of Boyacá, Infante and collaborators [12] carried out a morphological characterization of six varieties of quinoa, finding that adult plants have constant morphological characteristics, such as the presence of striae, pigmented axillae, and number of teeth on leaves. Morillo et al., 2020 [9] reported the existence of high morphological variability in 19 quinoa materials from the Department of Boyacá, where the more variable characteristics were the color of axillae and striae, plant height, number of panicles, seed yield per plant, and weight of 1000 grains.

The morphoagronomic characterization of quinoa accessions will facilitate the selection of materials that will improve production in regions with specific environmental conditions, initiate certified seed registration processes, discriminate accessions, determine potential uses, form core collections, identify duplicates in collections, and promote use in conservation and genetic improvement programs [9]. This study aimed to evaluate phenotypic characteristics with qualitative and quantitative descriptors of thirty accessions of *C. quinoa* to select outstanding accessions from the seed collection in the Department of Boyacá, Colombia.

## 2. Results

The climatic conditions of the study region were typical of tropical zones, characterized by few fluctuations in the photoperiod and average temperatures. The minimum temperature during this study ranged between 7.43 and 10.00 °C, and the maximum was between 17.80 and 20.00 °C, with an average temperature between 13.43 and 14.75 °C, and the average relative humidity was 78%. The daily illumination that the accessions received during the experiment was approximately 12 h.

### 2.1. Morphologic Characterization Using Quantitative Descriptors

The nine quantitative descriptors evaluated in the quinoa accessions had broad variation. The plant height had a mean of 176.7 cm, where the Quinua beteitiva accession exhibited the highest height (PH = 248.2 cm), and the largest panicle diameter (PD = 35.4 cm) (Table 1). However, the variables yield, seed weight, and seed diameter were zero because grains were not formed. In contrast, the accession with the lowest height was Quinua Peruana (PH = 111.9 cm), with PL = 39.0 cm, PD = 20.0 cm, NP = 6.8, and NT = 9.8, the lowest values for these variables, with respect to the other evaluated accessions. However, this accession presented the largest seed diameter (GD = 2.63 mm) and a higher-than-average yield value (Y = 62.02 g). On the other hand, Amarilla de maranganí had the highest weight of 1000 seeds with a value of 0.40 g and a seed diameter higher than average (GD = 2.57 mm). Quinua Blanca de Jericó Tuta2 presented the longest panicle length (PL = 72.4 cm) and plant height, 220.4 cm, but the yield was below average (Y = 18.17 g) (Table 1).

The Tukey test (*p* < 0.05) for all the quantitative variables, shows differences between the accessions evaluated. The best yields per quinoa plant were achieved by the accessions Quinoa primavera with 87.53 g, Quinoa real with 68.55 g, Quinoa semiamarga with 66.63 g, and Quinoa peruana with 62.02 g. The variables weight of the seeds (WS) and diameter of the seeds (GD) presented lower standard deviations (Table 1).

The analysis of variance detected statistically significant differences (*p* < 0.05) between the evaluated accessions for the characteristics plant height, stem diameter, length and diameter of the panicle, number of panicles, number of teeth on the leaf, yield, and seed weight and diameter.

The Spearman correlation analysis (*p* ≤ 0.05) between the quantitative variables showed that there were high and significant correlations between the weight (WS) and diameter (GD) of the seeds (r = 0.89), the length (PL) and height (PH) of the plants (r = 0.88), seed weight (WS) and yield (Y) (r = 0.82), and seed diameter (GD) and yield (Y) (r = 0.71). There were also negative correlations between yield (Y) and panicle length (PL) (r = −0.51), and yield (Y) and plant height (PH) (r = −0.50) (Figure 1).

The principal components analysis showed that 68.6% of the total variance was explained by the first two components (CP1 = 54.6% and CP2 = 14.0%) (Figure 2a). The variables that made the greatest contribution to the variation of CP1 included plant height, length, diameter, and number of panicles. The weight of 1000 seeds, seed diameter and stem diameter stem contributed more to the variation in CP2. On the other hand, the variables associated with weight and seed diameter correlated more with yield than the variables plant height, stem diameter, number of teeth on the leaves, and the variables associated with the panicle (Figure 2a).

The cluster analysis of the quantitative variables grouped the accessions into five clusters (Figure 2b). However, the groups were not established according to the collection area or place of origin. The first group had the accessions that presented the highest values for seed weight (0.32–0.40 g), seed diameter (2.16–2.63 mm) and yield (46.47–87.53 g). The second group had the Quinua beteitiva accession, which had the greatest height (PH = 248.2 cm) and panicle diameter (PD = 35.4 cm) but did not develop seeds. The accessions in the third group presented yields between 12.28 and 48.75 g, seed diameters from 1.87 to 2.36 mm, weights of 1000 seeds from 0.17 to 0.28 g, and stem diameters from 3.2 to 4.3 cm. The accessions in the fourth group included Quinua dulce de Tuta and Quinua semiamarga, with heights between 183.8 and 189.2 cm, panicle diameters between 2.2 and 2.3 mm, and yields between 55.49 and 66.63 g. Finally, group five had the accessions with the lowest values for yield (12.28 to 41.17 g), seed diameter (1.85 to 2.23 mm), and seed weight (0.16 to 0.25 g). These analyses were consistent with the principal component analysis.

These results were used to infer that the quantitative variables of yield, plant height and variables associated with the seeds, such as weight and diameter, were the most discriminative parameters, differentiating between the accessions evaluated in this study. Therefore, with these variables, it is possible to select materials to start breeding programs for quinoa.

### 2.2. Morphologic Characterization Using Qualitative Descriptors

When evaluating the qualitative variables, it was observed that, after germination and during the development of seedlings, the stem color was generally green, although, in some accessions, the color changed during flowering to shades of purple (Q. ceniza, Q. Sotaquirá, Tunkahuan—ICA, Amarilla de maranganí, Q. Peruana, Blanca de Jericó de Toca, Tunkahuan Tibasosa). For stem shape, all accessions had angular stems with striae (Figure 3a,b), which were green in 99% of the plants. Only three plants of the Quinua ceniza presented a purple color. Further, 20% of the evaluated plants had pigmented axillae (Figure 3c) that were purple in all cases (Tunkahuan, Amarilla de maranganí, Blanca dulce Soracá, Peruana, Quinua Siachoque, Piartal de Tibasosa, Blanca de Jericó de Toca, Quinua cremosa Malvinas). The seedlings had calcium oxalates that varied between white, pink, purple and purple (Figure 3h,i). The purple accessions included Quinua ceniza, Quinua colorada, Tunkahuan, and Quinua Siachoque. These oxalates were observed only until the flowering stage began and subsequently disappeared.

The leaves were green until physiological maturity. Then, in all accessions, they became yellow, starting from the basal leaves towards the apical ones, until senescence. The shape of the leaves had four types: lanceolate, rhomboidal, triangular, or oval, which is a polymorphic characteristic in the same plant. The most common forms were lanceolate in the apical branches, and triangular, oval, or rhomboidal in the basal leaves. (Figure 3g). The edge of the leaves was serrated, entire and dentate. In the basal leaves, 93% had serrated leaves, while in the apical leaves, 83% of the plants were serrated. The most common growth habit was branched up to the first third (61%), and a simple habit was observed at 37%.

The panicles demonstrated high variation for color that depended on the accessions and stage of development. Thus, the panicle colors observed at physiological maturity were: purple, pink, yellow, orange, red, green, and a mixture between these colors (Figure 4a–e). For panicle shape, 93% of the plants were glomerulate, while the intermediate and amarantiform forms were less common, appearing in only 7% of the plants. Panicle density was 61% loose, 29% intermediate, and 10% compact. Further, 95% of the plants had yellow flowers and 5% had white flowers.

There were four grain shape types: lenticular, cylindrical, ellipsoidal, and conical. Eighty three percent of the plants had a cylindrical grain shape, 7% were ellipsoidal, 6% were lenticular, and 4% were conical. For grain edge, 48% of the plants had a wavy edge, 26% were smooth, and 21% were intermediate (Figure 4g–i). The color of the episperm was also variable: 69% of the plants had a transparent episperm, 13% were white, 7% were black, and 6% were beige. The color of the perigonium presented orange, black, brown, and beige variations (Figure 4f). For seed germination capacity, good vigor was observed in 86% of the accessions. However, in accessions such as Quinua negra, the seeds required approximately 15 days for germination, while the average for the other accessions was 24 h.

The multiple correspondence analysis showed that 36.1% of the total variance was explained by the first two components, CP1 (19.0%) and CP2 (17.1%). The first component grouped the accessions according to characteristics such as fuchsia calcium oxalates, amarantiform panicle shapes, orange perigonia pink granules at bloom, and pink panicles at bloom. The second component was grouped according to transparent episperm, intermediate density, compact panicle, and white granules at flowering. Figure 5 shows the distribution of variables according to their contribution to the total variance in the first two components.

The cluster analysis of the qualitative variables formed seven groups (Figure 6). The first one had the accession Quinua beteitiva, which did not develop grains but developed panicles, which were lax and fragile. The second group was defined by orange panicles at flowering, purple granules at flowering, purple calcium oxalates, purple striae, and compact panicles. The third group was characterized by green panicles at bloom, green granules at bloom, white calcium oxalates, and intermediate panicle density. The fourth group was represented by Quinua negra, which had a conical grain shape and black perigonium and episperm. The fifth group had Quinua primavera and Quinua real, grouped by pink panicle at physiological maturity and intermediate panicles. Group six was characterized by green panicles at green bloom and wavy grain borders. Group seven contained 15 of the 30 accessions and did not present qualitative variables that defined the grouping, but most of the plants had a beige perigonium color, glomerulate panicle shape, white oxalate color, and lax panicle density.

In general terms, the qualitative variables that contributed significantly to the selection or discrimination of accessions included grain color and panicle density because of the presence and coloration characteristics of structures, such as axillae, striae, and panicles, that were highly variable between and within the evaluated quinoa accessions.

### 2.3. Morphologic Characterization Taking into Account the Joint Analysis of Qualitative and Quantitative Descriptors

The factorial analysis of mixed data considered all quantitative and qualitative variables and discriminated the accessions with outstanding morphologic characteristics. This analysis showed that the contribution of the variables to the first two components was 38.36%. The variables that contributed positively to CP1 (25.81%) included yield, weight, and diameter of the seeds (quantitative variables), yellow striae, white calcium oxalates, and purple axillae (qualitative variables) (Figure 7a). For CP2 (13.05%), the quantitative variables were yield and number of teeth on the leaves, and the qualitative variables were fuchsia calcium oxalates, uncolored axillae, and beige perigonia.

The cluster analysis formed nine groups (Figure 7b). The first group had the accessions Amarilla de Maranganí and Quinua peruana, which were characterized by below average seed weight and diameter (0.39 g and 2.59 mm), and panicle height and length (PH = 176.7 cm and PL = 57.9 cm). In the second group, ash quinoa was characterized by purple striae, grey perigonia and black episperm. Group three was characterized by plant heights between 175.4 to 202.9 cm, seed diameters from 2.06 to 2.23 mm, purple oxalates, and lax panicles. The fourth group had the accession Quinua beteitiva, which had a plant height that was higher than average, 248.2 cm, and did not produce grain. Group five was made up of the accessions Quinua Siachoque, Mezcla Siachoque 2, and Susunaga, which presented yields between 12.28 and 29.08 g of seeds per plant, weights of 1000 seeds from 0.16 to 0.23 g, and glomerulate, lax panicles. The accessions in group six were characterized by yellow striae, an average plant height of 181.4 cm, 10–15 panicles per plant, and seed diameters of 0.26 mm. Group 7 was made up of eleven accessions and did not present qualitative and quantitative variables that defined this grouping. However, these accessions had lower than average yields, that is, less than 37.16 g of seeds per plant, plant heights between 159.1 and 220.4 cm, and seed diameters from 1.87 to 2.36 mm. Quinua negra represented group 8, characterized by lower-than-average stem and seed diameters (3.1 cm and 1.7 mm), with a conical grain shape. Finally, group 9 was made up of the accessions Quinua primavera and Quinua real, which presented, on average, Y = 78.04 g of seeds per plant, PH = 130.9 cm, and PL = 42.9 cm.

The joint analysis of the morphologic descriptors proved to be robust and grouped the accessions according to the principal qualitative and quantitative variables, identifying promising accessions with the potential to start a breeding for quinoa in Colombia.

### 2.4. Promising Accessions Selection Index in Breeding Programs

The selection index confirmed the results obtained with the factorial analysis of mixed data, which established that the high-yield accessions were Quinua primavera, Quinua Peruana, Quinua real, and Amarilla de maranganí. These accessions were characterized by small sizes, large grains and good yield, which are ideal characteristics for the commercialization of quinoa grain in Colombia. The accessions Q. siachoque, Mezcla Siachoque 2, Blanca de Jericó Tuta2, and Q. beteitiva did not meet the needs of farmers because they had higher plant heights with no or little grain production (Figure 8).

## 3. Discussion

The quinoa from the seed collection of the Department of Boyacá, Colombia were phenotypically characterized to select elite accessions to improve quinoa production in the region. The life cycle of the evaluated quinoa accessions was approximately six months, with a photoperiod and average temperature that were constant in the study area. The yield had high variability between the accessions. In general, the accessions with plant heights higher than average (>176.7 cm) presented lower yields, as reported by [13]), who, when evaluating quinoa materials in southern Italy, found negative correlations between plant height and yield. [14] also observed a decrease in growth with the formation of grains in the Department of Cauca, Colombia in the Tunkahuan, Blanca de Jericó, and Blanca de Soracá varieties. This was probably due to the fact that the increase in plant growth coincides with the beginning of the reproductive stage, where the energy produced by plants is distributed in the growth and the beginning of panicles. Some genotypes use this energy only for growth, and grains do not develop properly because of the source–sink relationship [15,16].

It has been reported that longer panicles could provide higher grain yield than shorter ones [17,18]. However, this trend was not observed in the studied accessions. The Blanca de Jericó Tuta2 accession presented the longest panicle length, PL = 72.4 cm, and a below average yield (Y = 18.17 g). Quinua Peruana had the shortest panicle length, PL = 39.0 cm, and obtained an above average yield (Y = 62.02 g). This suggests that this descriptor could be useful in the selection of accessions with better yield. Additionally, the seed characteristics, such as weight and diameter, were correlated with yield. It should be noted that these two variables are important for the commercialization of quinoa and are commonly used as criteria for the selection of materials for the improvement of quinoa [19].

On the other hand, although qualitative variables constitute a fundamental tool to determine the adaptation strategies of plants and are used as varietal descriptors [20], in this study, these traits had broad genetic variability, as represented in the different colorations of the striae, axillae, panicles and seeds. In addition, these traits were highly variable within the same accessions, that is, there was high heterogeneity in the characteristics associated with these variables. This behavior was also observed in quinoa materials evaluated in the Rio Grande do Sul region of Brazil [21] and in cultivars of Quinua Blanca de Jericó, expressed in different pigmentations within individuals in structures such as panicles and stems. These variations allow plants to adapt more quickly to environmental conditions [22]. However, these variables are the basis for genetic improvement programs because, if there is no variability, no selection can be made, since all individuals respond in the same way to the evaluated conditions. Therefore, the existence of phenotypic variability associated with qualitative or quantitative morphological characteristics will allow the selection of materials that respond to the needs of farmers, producers, and consumers.

The joint analysis of the quantitative and qualitative variables differentiated the accessions with higher yields and identified promising genetic lines. The groupings of the accessions Quinua amarilla de maranganí, Quinua peruana, Quinua primavera, and Quinua real remained constant in all analyses, meaning they have high potential for the extensive production of quinoa grain because their yields are higher and they present stable morphological characteristics, such as grain color and panicle density. For phenotyping quinoa with important agronomic traits, the quantitative variables that should be considered are yield, plant height, stem diameter, panicle length, weight of 1000 seeds, and grain diameter, while qualitative variables are panicle density and grain color since they are useful for the selection of materials with potential for quinoa production.

The broad variability in the morphological traits observed in this study may have been due to the facts that farmers maintain mixtures of different materials in the same crop. In Colombia, selection parameters have not been defined for quinoa materials because of a lack of knowledge. Therefore, farmers have marketing difficulties because of a loss of grain quality and decreases in production caused by a lack of selection of planting material and pure materials. Thus, these findings regarding the high variability in both qualitative and quantitative descriptors suggest that they can be very useful in breeding programs. In addition to increasing the productivity of crops, it is possible for accessions to demonstrate great capacity to adapt to different environmental conditions, because the evaluated materials have the advantage of agroecological adaptation to the region since they have been cultivated in the region [23].

However, there are additional morphological characteristics for the good commercialization of the quinoa grain in Colombia, such as small size of plant, erectness, and uniformity, which facilitate the harvesting process, a single terminal panicle with compact glomeruli, and large, white grains [9,24]. Therefore, the proposed selection index confirmed the outstanding accessions (Quinua primavera, Quinua Peruana, Quinua real, and Amarilla de maranganí) using these descriptors since these characteristics are decisive when estimating the commercial quality of crops, with the advantage that these accessions have already been cultivated in the environmental conditions of the region.

Finally, our results suggest that, for Colombia, it is essential to continue with the characterizations of quinoa accessions using morphological descriptors and to include biochemical and molecular descriptors because of the large number of mixtures present in cultivars, allowing efficient selection if pollination is controlled and accessions that exhibit undesirable characteristics to be eliminated, thereby obtaining a variety that responds to the needs of farmers, producers, and consumers through the development and implementation of adequate breeding schemes.

## 4. Materials and Methods

A total of 30 accessions of quinoa (*C. quinoa*) were evaluated, which belong to the seed collection of the Department of Boyacá (Table 2). The morphoagronomic characterization was carried out under greenhouse conditions in the city of Tunja, located at an altitude of 2690 m.a.s.l., with an average temperature of 13 °C, relative humidity of 78%, and a 12:12 photoperiod. The germination of the seeds was carried out in the nursery with a mixture of humus and peat in a 2:1 ratio. By accessions, 16 alveoli were sown whereby three seeds were placed that were taken randomly, after 20 days of growth when the seedlings had six true leaves. They were transplanted to the greenhouse beds, and thinning was carried out when more than two plants grew per alveolus. The accessions were sown under a randomized complete block design (RCB) of three plants per block (three blocks) for a total of nine repetitions for each accession, with conventional agronomic management. The harvest was carried out manually when the plants reached physiological maturity.

Twenty-one morphologic descriptors were evaluated, of which nine were quantitative and twelve qualitative, defined by the FAO for quinoa [6] (Table 3). Measurements were taken on nine individuals of each accession.

For the selection of accessions with important and highly productive agronomic characteristics, the program R was used to analyze selection indices (RIndSel: R software to analyze Selection Indices) to obtain Smith’s linear phenotypic selection index [25]. The following variables were weighted: yield, plant height, seed diameter and grain color, as described below:I.S = *Yield (0.95) − Plant height (0.94) + Grain diameter (3.70) + Grain color (1.31).*

The variables yield, grain diameter, and grain color were expressed positively since plants with more grams of seeds per plant, a greater seed diameter and light grain colors are sought. The plant height was expressed negatively since low-bearing accessions are sought.

### Statistical Analysis

For the quantitative variables, a descriptive analysis was carried out. Then, the assumptions for the parametric analyzes were verified, and the analysis of variance (ANOVA) was carried out. To determine the significant differences between treatments, a Tukey multiple comparison test was performed with *p* < 0.05. These analyses were performed using R Core Team [26] and the missMDA package [27]. The Spearman correlation was estimated and plotted using the R package “corrplot”: Visualization of a Correlation Matrix (Version 0.84) [28]. For the multivariate analysis, a hierarchical grouping with principal components (HCPC) was carried out with the algorithms in the factoextra package of the R program [29], which were plotted on a two-dimensional plane using the FactoMineR package [30]. The dendrogram was done using the main components, Euclidean distance, and Ward’s minimum variance hierarchical grouping method with the FactoMineR package [30] in the R program [26].

For the qualitative variables, frequency analyses were performed with Infostat [31], a multiple correspondence analysis with the algorithms in R program’s factoextra package [29], and the dendrogram obtained with the components, Euclidean distance, and Ward’s minimum variance hierarchical grouping method, using the FactoMineR package [30] in the R program [26]. For the joint analysis of the quantitative and qualitative variables, a factorial analysis of mixed data was carried out with the factoextra package in the R program. Additionally, a dendrogram was generated using the Euclidean distance and hierarchical grouping method of Ward’s minimum variance with the FactoMineR package [30]. Finally, the selection index was calculated with an accessions percentage of 5% and the variance-covariance matrix for the variables yield, plant height, seed diameter, and grain color using the RindSel program. The index for each of the accession was plotted in Microsoft Excel 2013.

## 5. Conclusions

The characteristics, such as seed diameter, panicle density, plant height, and grain color, allowed for the selection of quinoa accessions with better yield and desirable agronomic characteristics. The broad phenotypic variability in the accessions, in terms of both grain quality and yield, constitutes a fundamental tool for recording varieties that improve the positioning of quinoa in Colombia. In addition, this study revealed relationships between yield and morphological characteristics, which could be useful for the selection of parental lines in future breeding programs that seek to generate quinoa hybrids in Colombia. Our results indicated that the accessions Quinua primavera, Quinua Peruana, Quinua real, and Amarilla de maranganí represent the most promising parents for the future development of breeding programs that aim to respond to the needs of farmers, producers, and consumers in Colombia.

## Figures and Tables

**Figure 1 plants-10-01339-f001:**
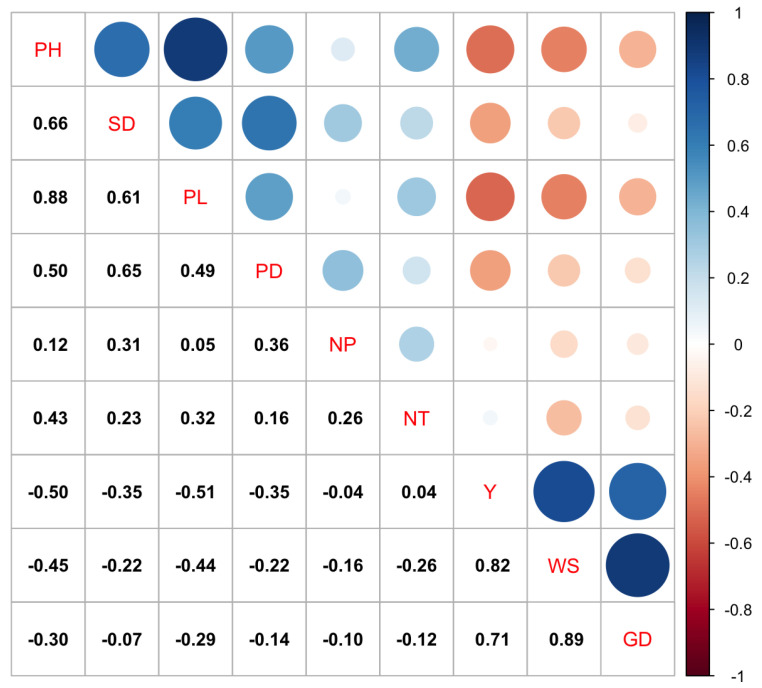
Spearman correlation analysis among the quantitative variables in the 30 quinoa accessions.

**Figure 2 plants-10-01339-f002:**
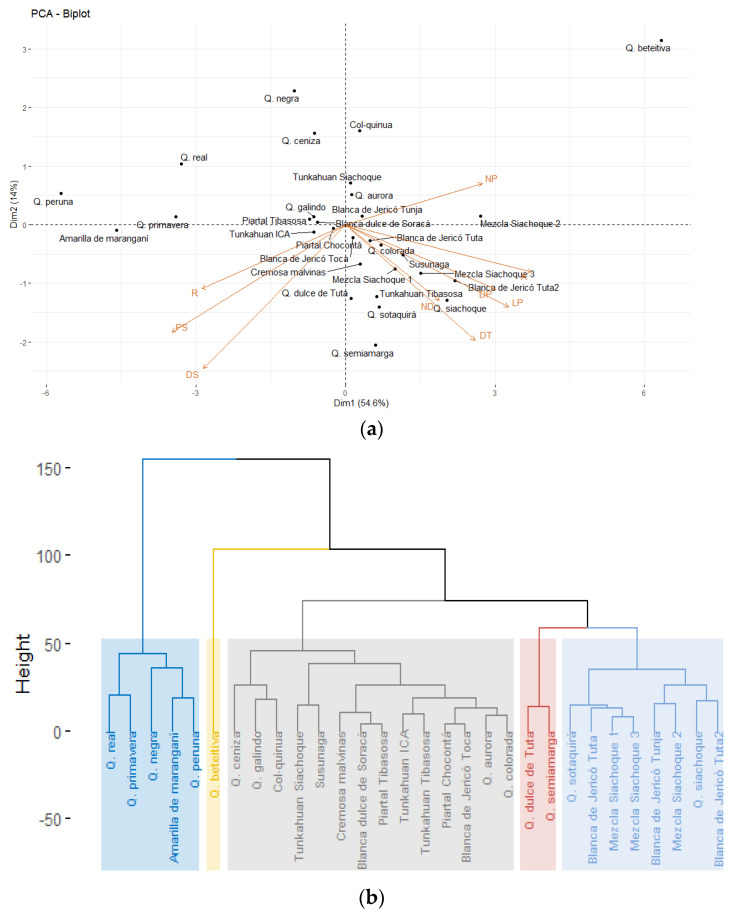
(**a**) Principal component analysis biplot. The variables that contributed to yield were weight and diameter of the seeds; (**b**) Hierarchical cluster analysis of quinoa accessions considering quantitative variables.

**Figure 3 plants-10-01339-f003:**
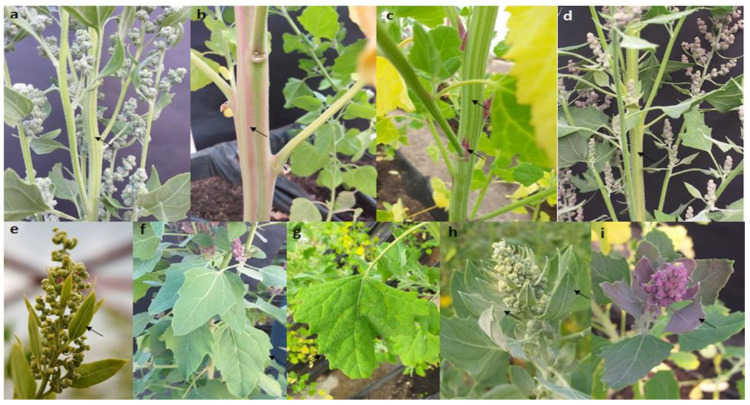
(**a**) Angled stem with green striae; (**b**) Purple striae; (**c**) Stem with green striae and purple axillae; (**d**) Stem with green striae and no axillae; (**e**) Apical leaves with lanceolate and entire shape; (**f**) Leaves with rhomboid shapes and serrated edge; (**g**) Triangular shaped leaves with serrated edge; (**h**) White calcium oxalates on the upper surface and underside of the leaf; (**i**) Pink calcium oxalates on the upper surface of the leaf.

**Figure 4 plants-10-01339-f004:**
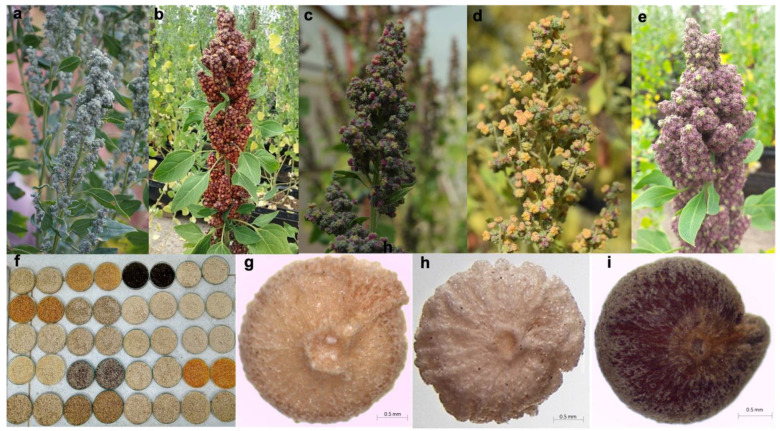
(**a**) Lax green panicles; (**b**) Compact, amarantiform, and purple; (**c**) Loose, and green and purple; (**d**) Lax with a mix of green and yellow; (**e**) Compact purple panicle; (**f**) Color of the seed episperm; (**g**) Seed with a beige intermediate border; (**h**) Seed with beige wavy edge; (**i**) Smooth edge seed with brown perigonium.

**Figure 5 plants-10-01339-f005:**
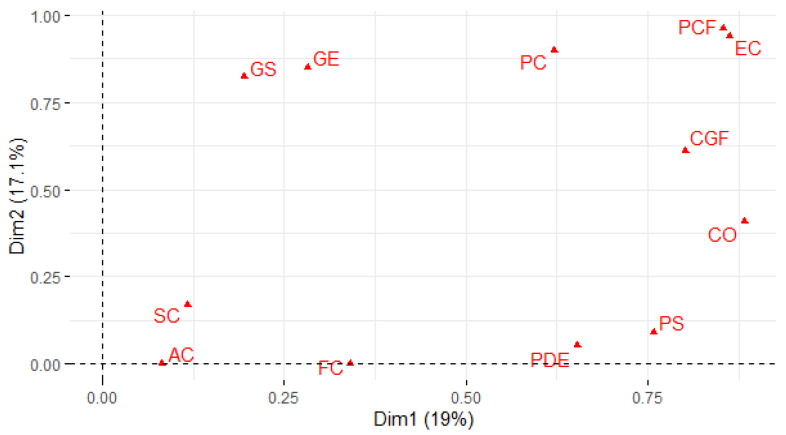
Multiple correspondence analysis showing the contribution of variables, ordering the accessions according to the qualitative variables.

**Figure 6 plants-10-01339-f006:**
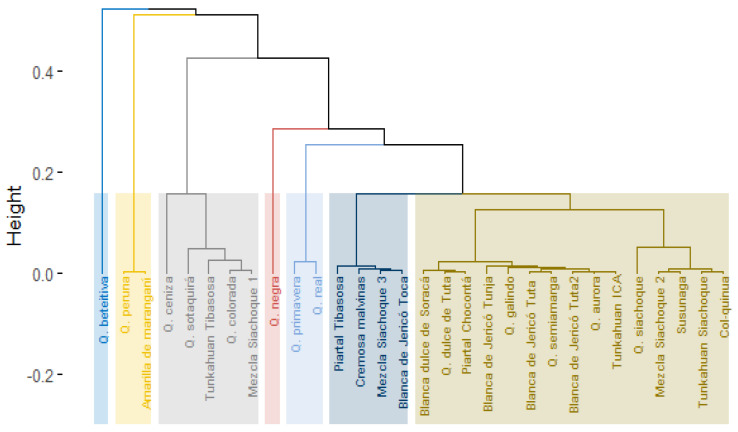
Cluster analysis showing seven groups of quinoa accessions formed according to the qualitative variables.

**Figure 7 plants-10-01339-f007:**
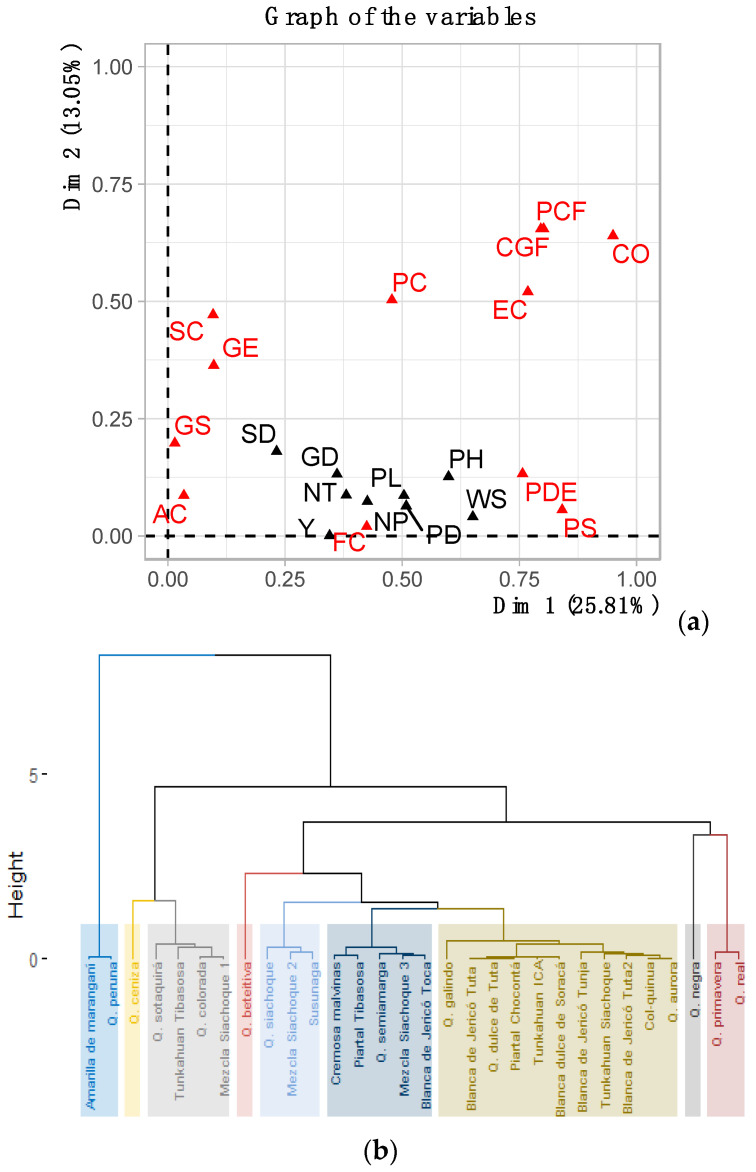
(**a**) Mixed factor analysis of the contribution of the variables, ordering the accessions according to the qualitative and quantitative variable; (**b**) Cluster analysis, showing nine groups of accessions formed according to the qualitative and quantitative variables.

**Figure 8 plants-10-01339-f008:**
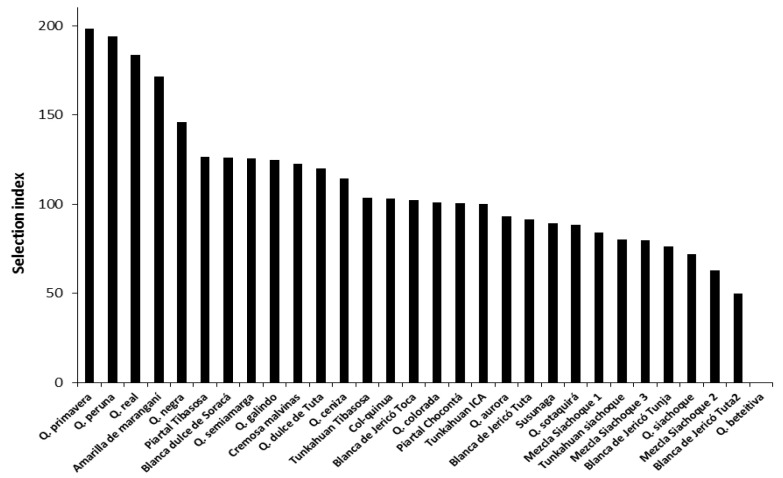
Values of the selection index for each of the evaluated quinoa accessions (*C. quinoa*), showing the discrimination of accessions according to agronomic characteristics.

**Table 1 plants-10-01339-t001:** Averages, standard deviation, and Tukey multiple comparison (MC) for the quantitative variables of the thirty quinoa accessions.

Accessions	PH (cm) (S.D.)	SD (cm) (S.D.)	PL (cm) (S.D.)	PD (cm) (S.D.)	NP (#) (S.D.)	NT (#) (S.D.)	Y (g) (S.D.)	WS (g) (S.D.)	GD (mm) (S.D.)
*Quinoa real*	128.6 (14.35) g, h, j	3.2 (0.62) b, d	43.6 (10.73) a, c, e, g	26.6 (7.06) a, b, c, d, e	12.3 (2.35) a, b, c, d, e	13.8 (6.82) a, b, d	68.55 (37.62) b e	0.32 (0.02) f	2.16 (0.09) a e g
*Quinoa aurora*	182.9 (23.60) a, b, d, f, i	3.5 (0.49) a, b, c, d	57.7 (12.14) a, b, c, d, e, f, g	30.2 (2.91) a, b, c, d, e	12.7 (2.55) b, c, d, e	17.3 (4.12) a, b, c, d	27.08 (17.55) a d f g i	0.23 (0.03) a b d g h j	2.05 (0.08) b c f g
*Quinoa ceniza*	148.6 (28.80) a, c, g, h, j, k	3.2 (0.44) b, d	56.8 (20.42) a, b, c, d, e, f, g	24.8 (4.97) a, c, e	11.9 (3.79) a, b, c, e	21.6 (5.46) a, c	19.40 (9.48) d f g i	0.21 (0.03) a c g i j	1.87 (0.14) b c f h
*Quinoa beteitiva*	**248.2 (11.29) e**	4.0 (0.50) a, c, d	71.6 (13.64) b, d	**35.4 (4.88) d**	17.7 (5.10) b, d	17.1 (6.57) a, b, c, d	**0.00**	**0.00**	**0.00**
*Quinoa sotaquirá*	202.9 (25.17) b, f, i	3.9 (0.62) a, b, c, d	64.9 (13.33) b, d, f	31.7 (8.49) b, c, d, e	11.2 (4.63) a, c, e	20.0 (5.57) a, c, d	41.17 (20.70) a b c d f g h i	0.25 (0.03) a b d g h	2.23 (0.09) a e g
*Quinoa negra*	138.7 (18.99) c, g, h, j, k	**3.1 (0.35) b**	40.2 (8.80) e, g	27.4 (7.75) a, b, c, d, e	13.8 (3.27) b, d, e	21.8 (8.15) a, c	44.34 (20.63) a	0.16 (0.01) i	1.75 (0.05) h
*Tunkahuan ICA*	186.3 (13.25) a, b, d, f, i	3.6 (0.32) a, b, c, d	52.3 (6.04) a, b, c, d, e, f, g	27.1 (5.69) a, b, c, d, e	12.0 (2.12) a, b, c, e	20.3 (4.74) a, c, d	36.73 (18.45) a b c d f g h i	0.27 (0.06) b d f h	2.21 (0.07) a e g
*Blanca de Jericó Tuta*	192.2 (22.92) b, d, f, i	3.7 (0.48) a, b, c, d	64.2 (14.33) b, d, f	25.1 (7.67) a, b, c, e	13.3 (2.65) b, d, e	22.2 (8.09) a, c	34.00 (11.09) a c d f g i	0.22 (0.03) a c d g h j	2.15 (0.11) a e g
*Amarilla de maranganí*	120.3 (11.12) g, j	3.6 (0.49) a, b, c, d	41.1 (7.18) a, e, g	22.4 (6.56) a, c	7.2 (2.17) a, c	12.0 (9.27) b, d	46.46 (14.49) a b c f h i	**0.40 (0.02) e**	2.57 (0.10) d
*Quinoa colorada*	175.4 (18.60) a, c, d, f, i	3.7 (0.42) a, b, c, d	59.6 (6.65) a, b, c, d, f, g	33.8 (4.99) b, d, e	13.1 (2.57) b, d, e	19.3 (4.58) a, c, d	26.89 (19.12) a d f g i	0.23 (0.03) a b d g h j	2.17 (0.14) a e g
*Blanca dulce de Soracá*	169.2 (25.57) a, c, d, f, h, i, k	3.5 (0.62) a, b, c, d	51.4 (10.79) a, b, c, e, f, g	30.0 (7.81) a, b, c, d, e	12.9 (2.80) b, c, d, e	20.4 (5.08) a, c, d	47.53 (24.32) a b c f h i	0.24 (0.03) a b d g h	2.14 (0.16) a g
*Piartal Chocotá*	183.0 (20.11) a, b, d, f, i	3.5 (0.36) a, b, c, d	64.7 (7.58) b, d, f	29.1 (5.53) a, b, c, d, e	11.1 (3.86) a, c, e	16.9 (7.75) a, b, c, d	34.42 (16.09) a c d f g h i	0.25 (0.02) a b d g h	2.16 (0.07) a e g
*Quinoa dulce de Tuta*	183.8 (20.25) a, b, d, f, i	3.8 (0.62) a, b, c, d	65.2 (12.85) b, d, f	28.8 (5.09) a, b, c, d, e	13.3 (3.91) b, d, e	22.4 (7.54) a, c	55.49 (24.48) a b c e h	0.26 (0.03) b d g h	2.28 (0.20) a e
*Quinoa semiamarga*	189.2 (14.53) a, b, d, f, i	4.0 (0.49) a, c, d	61.1 (9.36) a, b, c, d, f	33.0 (5.41) b, d, e	14.0 (3.84) b, d, e	**24.2 (3.53) c**	66.63 (27.87) b e h	0.27 (0.02) b f h	2.24 (0.05) a e g
*Quinoa peruana*	**111.9 (16.17) j**	3.3 (0.38) a, b, d	**39.0 (6.86) e**	**20.0 (4.66) a**	**6.8 (2.39) a**	9.8 (7.90) b	62.02 (16.69) b c e h	0.39 (0.01) e	**2.63 (0.09) d**
*Quinoa siachoque*	207.9 (25.78) b, e, i	3.9 (0.54) a, b, c, d	70.8 (7.36) b, d	34.7 (7.94) b, d, e	14.6 (4.25) b, d, e	20.7 (6.40) a, c, d	29.08 (18.02) a d f g i	0.23 (0.02) a b d g h j	2.18 (0.09) a e g
*Blanca de Jericó Tuta2*	220.4 (32.14) b, e	3.8 (0.42) a, b, c, d	**72.4 (14.98) d**	33.6 (2.74) b, d, e	12.1 (3.82) a, b, c, d, e	20.7 (3.46) a, c, d	18.17 (9.63) d f g i	0.20 (0.03) a c i j	2.08 (0.08) a f g
*Piartal Tibasosa*	169.8 (32.82) a, c, d, f, i, k	3.5 (0.45) a, b, c, d	53.3 (10.58) a, b, c, d, e, f, g	29.0 (6.80) a, b, c, d, e	14.4 (3.28) b, d, e	18.2 (3.67) a, b, c, d	48.75 (17.54) a b c f h	0.27 (0.02) b d f h	2.21 (0.06) a e g
*Blanca de Jericó Tunja*	196.6 (36.14) b, d, f, i	3.6 (0.36) a, b, c, d	71.1 (15.38) b, d	26.3 (5.52) a, b, c, d, e	11.1 (2.93) a, c, e	15.3 (5.00) a, b, c, d	23.07 (14.00) d f g i	0.22 (0.03) a c d g j	2.12 (0.08) a g
*Blanca de Jericó Toca*	179.9 (24.10) a, b, d, f, i	3.6 (0.30) a, b, c, d	65.3 (7.16) b, d, f	30.4 (7.95) b, c, d, e	12.6 (2.46) a, b, c, d, e	16.9 (7.94) a, b, c, d	33.48 (12.47) a c d f g i	0.25 (0.02) a b d g h	2.13 (0.11) a g
*Cremosa malvinas*	169.8 (18.64) a, c, d, f, i, k	3.9 (0.50) a, b, c, d	60.3 (12.31) a, b, c, d, f, g	31.0 (4.44) b, c, d, e	15.2 (5.19) b, d, e	18.2 (5.87) a, b, c, d	44.64 (23.46) a b c f g h i	0.27 (0.02) b d f h	2.17 (0.06) a e g
*Tunkahuan Tibasosa*	187.0 (18.30) a, b, d, f, i	4.1 (0.37) a, c	57.8 (10.03) a, b, c, d, e, f, g	32.4 (4.88) b, c, d, e	14.0 (1.66) b, d, e	19.1 (8.43) a, c, d	41.59 (23.84) a b c d f g h i	0.28 (0.02) b f	2.17 (0.06) a e g
*Tunkahuan siachoque*	185.1 (46.69) a, b, d, f, i	3.6 (0.89) a, b, c, d	60.9 (20.03) a, b, c, d, f	28.3 (8.06) a, b, c, d, e	11.3 (5.70) a, c, e	14.0 (8.54) a, b, d	15.44 (14.15) d g i	0.22 (0.03) a c d g j	2.06 (0.11) b f g
*Mezcla Siachoque 1*	194.9 (21.17) b, d, f, i	3.8 (0.46) a, b, c, d	62.9 (8.88) b, c, d, f	31.4 (4.59) b, c, d, e	11.3 (3.28) a, c, e	22.9 (9.01) a, c	29.60 (22.49) a d f g i	0.22 (0.03) a c d g h j	2.06 (0.07) b c f g
*Mezcla Siachoque 2*	200.1 (32.27) b, d, f, i	**4.3 (0.43) c**	64.9 (13.12) b, d, f	27.6 (7.45) a, b, c, d, e	**17.9 (6.97) d**	19.1 (6.33) a, c, d	12.63 (13.85) d g	0.16 (0.05) c i	1.87 (0.20) b c h
*Mezcla Siachoque 3*	198.4 (19.14) b, d, f, i	4.1 (0.40) a, c	67.7 (9.75) b, d, f	33.2 (7.14) b, d, e	10.9 (4.20) a, c, e	17.6 (5.73) a, b, c, d	28.85 (17.68) a d f g i	0.22 (0.04) a c d g j	1.85 (0.12) c h
*Quinoa primavera*	133.3 (17.87) g, h, j, k	3.5 (0.67) a, b, c, d	42.3 (5.15) a, e, g	24.4 (6.39) a, c, e	12.3 (3.00) a, b, c, d, e	16.2 (6.36) a, b, c, d	**87.53 (40.89) e**	0.32 (0.04) f	2.24 (0.15) a e g
*Quinoa Galindo*	159.1 (21.00) a, c, d, g, h, k	3.7 (0.38) a, b, c, d	47.8 (6.55) a, c, e, f, g	35.2 (5.29) b, d	15.0 (2.00) b, d, e	11.8 (4.74) b, d	35.63 (16.16) a c d f g h i	0.28 (0.02) b f	2.36 (0.09) e
*Col-quinua*	165.8 (34.01) a, c, d, f, h, k	3.7 (0.52) a, b, c, d	47.6 (17.83) a, c, e, f, g	28.9 (8.99) a, b, c, d, e	14.7 (1.94) b, d, e	15.8 (5.33) a, b, c, d	21.46 (12.66) d f g i	0.17 (0.06) c i j	1.87 (0.23) b c h
*Susunaga*	172.2 (24.10) a, c, d, f, i, k	4.3 (0.66) c	59.8 (15.44) a, b, c, d, f, g	33.9 (7.77) b, d, e	11.4 (3.43) a, c, e	15.1 (4.26) a, b, c, d	12.28 (9.21) d	0.21 (0.04) a c g i j	2.08 (0.15) a g
**Mean**	176.7	3.7	57.9	29.5	12.7	18.0	38.20	0.25	2.14

PH Plant height, SD Stem diameter, PL Panicle length, PD Panicle diameter, NP N ° plant panicles, NT N ° teeth per leaf, Y Yield per plant, WS Weight of 1000 seeds, GD Grain diameter. Averages in each column with the same letters do not differ statistically (Tukey *p* < 0.05). The maximum and minimum values for each variable are in bold.

**Table 2 plants-10-01339-t002:** Sites of origin of the evaluated quinoa (*C. quinoa*) accessions.

	Accessions	Location	Coordinates
1	*Quinoa real*	Ventaquemada	5°22′00.4″ N 73°31′16.9″ W
2	*Quinoa aurora*	Soracá	5°30′06.9″ N 73°20′00.5″ W
3	*Quinoa ceniza*	La colorada Tunja	5°34′44.7″ N 73°20′36.0″ W
4	*Quinoa beteitiva*	Beteitiva	5°54′39.1″ N 72°48′31.2″ W
5	*Quinoa sotaquirá*	Sotaquirá. Vereda Bociga	5°45′57.6″ N 73°14′52.2″ W
6	*Quinoa negra*	La colorada Tunja	5°34′44.7″ N 73°20′36.0″ W
7	*Tunkahuan ICA*	ICA Surbatá	5°47′45.5″ N 73°04′20.2″ W
8	*Blanca de Jericó Tuta*	Tuta	5°41′26.6″ N 73°13′39.1″ W
9	*Amarilla de maranganí*	Pasca	4°18′32.8″ N 74°17′59.6″ W
10	*Quinoa colorada*	La colorada Tunja	5°34′44.7″ N 73°20′36.0″ W
11	*Blanca dulce de Soracá*	Soracá	5°30′06.9″ N 73°20′00.5″ W
12	*Piartal Chocotá*	Chocontá	5°08′44.3″ N 73°41′07.0″ W
13	*Quinoa dulce de Tuta*	Tuta	5°41′26.6″ N 73°13′39.1″ W
14	*Quinoa semiamarga*	Duitama	5°49′36.3″ N 73°02′03.9″ W
15	*Quinoa peruna*	Cómbita	5°38′01.9″ N 73°19′28.4″ W
16	*Quinoa siachoque*	Siachoque	5°30′45.3″ N 73°14′44.3″ W
17	*Blanca de Jericó Tuta2*	Tuta	5°41′26.6″ N 73°13′39.1″ W
18	*Piartal Tibasosa*	Tibasosa	5°44′40″ N 73°14′16″ W
19	*Blanca de Jericó Tunja*	Tunja	5°31′4″ N 73°23′48″ W
20	*Blanca de Jericó Toca*	Toca-Vda Tuaneca	5°34′03.6″ N 73°11′24.2″ W
21	*Cremosa malvinas*	Siachoque	5°31′00.8″ N 73°14′59.7″ W
22	*Tunkahuan Tibasosa*	Tibasosa	5°44′40″ N 73°14′16″ W
23	*Tunkahuan siachoque*	Siachoque-Finca San Antonio	5°31′55.6″ N 73°16′10.6″ W
24	*Mezcla Siachoque 1*	Siachoque	5°31′00.8″ N 73°14′59.7″ W
25	*Mezcla Siachoque 2*	Siachoque	5°31′00.8″ N 73°14′59.7″ W
26	*Mezcla Siachoque 3*	Siachoque	5°31′00.8″ N 73°14′59.7″ W
27	*Quinoa primavera*	Siachoque-Sabana de Bogotá	4°24′56.3″ N 74°06′06.0″ W
28	*Quinoa galindo*	Cómbita	5°38′01.9″ N 73°19′28.4″ W
29	*Col-quinua*	Cómbita	5°38′01.9″ N 73°19′28.4″ W
30	*Susunaga*	Cómbita	5°38′01.9″ N 73°19′28.4″ W

**Table 3 plants-10-01339-t003:** Morphologic descriptors used for the characterization of quinoa accessions from the Department of Boyacá.

Qualitative	Acronyms	Quantitative	Acronyms	Unit of Measurement
Calcium oxalates color	(CO)	Plant height	(PH)	cm
Strie color	(SC)	Stem diameter	(SD)	cm
Axil color	(AC)	N ° teeth per leaf	(NT)	#
Color of the granules at flowering	(CGF)	Panicle length	(PL)	cm
Panicle color at flowering	(PCF)	Panicle diameter	(PD)	cm
Flower color	(FC)	N ° plant panicles	(NP)	#
Panicle shape	(PS)	Yield per plant	(Y)	g
Panicle density	(PDE)	Weight of 1000 seeds	(WS)	g
Grain shape	(GS)	Grain diameter	(GD)	mm
Episperm color	(EC)			
Grain edge	(GE)			
Perigonium color	(PC)			

## Data Availability

The data generated within this work are open access and available to be shared with interested persons.

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
