# Peer review of "Phenotypic Characterization of Quinoa (Chenopodium quinoa Willd.) for the Selection of Promising Materials for Breeding Programs"

_plants, 2021, doi:10.3390/plants10071339_

Round 1
Reviewer 1 Report
Comments.
Interesting and useful manuscript, in the sense of characterizing quinoa genotypes, phenotyping that can be used in a series of research activities including the breeding area. However, must recognize the limitations of using data for the target region because it is cultivated under controlled conditions. This is because the environmental data in the greenhouse, as well as temperature, luminosity, precipitation, among others, are different from the region cultivated by producers and because the root system is in a pot with limited volume. For this reason, the word morpho agronomic analyzes could be replaced by morphological analisys. Similarly, yield that refers to grain yield at one spacing and density under field conditions could be replaced by weight of grain per plant.
Line20: replace morph agronomic for morphologic.
Line 23: Use Italic for C. Quinoa
L23. Replace yield for weight of grains per plant.
L29. higher grain weight per plant (yield is not apropriate because is a experiment in greenhouse condition.
L33. I would use only morphological. Morpho agronomic traits for release cultivars may be different under field and controled conditions.
L34. Adicional Key word. Pseudocereal.
L51. Collection of quinoa germoplasm. any bibliographic citation?
L75. Improvement programs, although depending on the type of substrate, the quality of the light and the size of the pots, there may be phenotypic differences between materials grown in a greenhouse and in field conditions.
L75. This.....
L84. Monthly precipitation of 2 mm? In the greenhouse? Precipitation or irrigation?
L89. Replace yield for grain weight per plant
L89. Comment. Considering the high coeficient of variation, In the discussion, put that this was a factor that limited the experiment.
L92. Replace yield for grain weight per plant.
L97. Value (y=62.02g0. This statement gives the impression that 62.02 is the average value.
L99. Similarly, this statement give us a impression that 2,57 is the average value.
L103. Adjust formationg in the title of table 3.
Table 3. How are the statistical differences among genotypes?
L106. Adjust formating.
L108. Are in bold. Statistical analysis would be better.
Df 2 in the table four means that rthe eperiment had 3 repetitions.
L117. Adjust formating
L128. indicate the correlation code in the text in figure 1.
L146. Figures 2 a and b are too small. Increase the size of these two figures. You can divide in figure 2 and figure 3 or increase size both figures. It is difficult to vuisualize this figure. Increase the size of it.
L146. it is difficult to check these data in the figure 2a and 2b.
L162. Made it instead of male it.
L164. Morphological instead of morphoagronomic.
L171. Twenty per cent of... instead of 20%.
L176. Replace process for stage.
P198. Ninety five per cent instead of 97%
L.209. Eight per cent instead of 80%
L222. increase size of fig. 5
L244. Morphological instead of morpho agronomic
L250. Weight of grains per plant instead of yield.
L252. Grain weight per plant
L258. Grey instead of gray Iguess.
L282. Morphological instead of morpho agronomic.
L291. grain weight per plant instead of yield.
L302. Discuss if the temperature, light and humidity conditions inside the greenhouse and the target environment are similar.
L307. Grain weight of grain per plant replacing yield.
L307. In general, the materials...
L315. And...
L319. Replace behavioura for trend.
L334. Allow instead of allows.
L349. Replace by morphological
L360. Replace morpho agaronomicc for morphological.
L369. Replace morf. agronomic for morphological
L375. I still prefer MM before results.
L381. Measurements inside greenhouse?
L382. According to Analisys of variance (df) (Table 4) you had 3 repetitions. You may have sub samples in the same pot.
L384. grain weight at what grain moisture?
L386. Adjust text position on the table.
Table 1. Adjust coordinates position on the table.
L.387. Twenty one...instead of 21
L390. Adjust formatting in title and at the bottom of tables.
L395. any color scale?
L428. better weight of grains per plant and hopfully yield in the field
Suggestions directly in the manuscrit can be found in a file included in the review.

Author Response
Response to Reviewer 1 Comments
“Phenotypic Characterization of Quinoa (Chenopodium quinoa Willd.) for the Selection of Promising Materials for Breeding Programs” by: Elsa Helena Manjarres-Hernández, Diana Marcela Arias Moreno, Ana Cruz Morillo-Coronado, Zaida Zarely Ojeda-Pérez, Agobardo Cárdenas-Chaparro.
- Interesting and useful manuscript, in the sense of characterizing quinoa genotypes, phenotyping that can be used in a series of research activities including the breeding area. However, must recognize the limitations of using data for the target region because it is cultivated under controlled conditions. This is because the environmental data in the greenhouse, as well as temperature, luminosity, precipitation, among others, are different from the region cultivated by producers and because the root system is in a pot with limited volume. For this reason, the word morpho agronomic analyzes could be replaced by morphological analisys. Similarly, yield that refers to grain yield at one spacing and density under field conditions could be replaced by weight of grain per plant.
A1: Thanks for your suggestion. In this new version of the R1 manuscript, we replace the morphoagronomic analysis with morphological analysis. The yield is presented throughout the manuscript as grain weight per plant, which represents the yield obtained per plant. We use this way of estimating yield based on research on morphological characterization, characterization of yield components, or genotype selection in quinoa that calculate yield as grain weight per plant. For example, Al-Naggar et al., 2017 when evaluating the effects of drought stress on morphological, physiological and yield characteristics, presents the yield measured as seed / plant yield (SYPP) in grams measured as the weight of seeds. per floor (on 10 floors). In the work of Mestanza et al., 2019, the agronomic characteristics of 21 genotypes of quinoa from Bolivia were analyzed and the yield was presented as grams of seeds per plant, finally presenting the O-5 genotype with 143.15 g. plant-1 as the material with the highest yield per plant (g).
- Al-Naggar, M., Abd El-Salam, R., Badran, A., and El-Moghazi, M. 2017. Genotype and Drought Effects on Morphological, Physiological and Yield Traits of Quinoa (Chenopodium quinoa ) Asian Journal of Advances in Agricultural Research 3(1): 1-15. Doi: 10.9734/ajaar/2017/36655
- Mestanza, C., Zambrano, K., Pinargote, J., Veliz, D., Vásconez, G., Fernández, N., and Olmos, E. 2019. Evaluación agronómica de genotipos de quinua (Chenopodium quinoa) en condiciones agroclimáticas en la zona de Mocache. Agricultural Sciences 12(1): 19-30. Doi: 10.18779/cyt.v12i1.299
- Line20: replace morph agronomic for morphologic.
A2: Thanks for your observation. In this new version R1, we realized this correction.
- Line 23: Use Italic for Quinoa
A3: Thanks for your observation. In this new version R1, we realized this correction.
- Replace yield for weight of grains per plant.
A4: Thanks for your suggestion.
- higher grain weight per plant (yield is not apropriate because is a experiment in greenhouse condition.
A5: Thanks for your suggestion. We mentioned that the materials present good yields in terms of grain weight per plant.
- I would use only morphological. Morpho agronomic traits for release cultivars may be different under field and controled conditions.
A6: Thanks for your observation. In this new version R1, we realized this correction.
- Adicional Key word. Pseudocereal.
A7: Thanks for your suggestion. In this new version R1, we add pseudocereal as a keyword.
- Collection of quinoa germoplasm. any bibliographic citation?
A8: Thanks for your observation. We add the bibliography of Morillo et al. 2020 who made a first approach to this collection of quinoa germplasm.
- Morillo, A., Manjarres, E., and Morillo, Y. 2020. Evaluación morfoagronómica de 19 materiales de Chenopodium quinoa en el Departamento de Boyacá. Biotecnología en el Sector Agropecuario y Agroindustrial 18, 84–96. https://doi.org/10.18684/bsaa.v18n1.1416
- Improvement programs, although depending on the type of substrate, the quality of the light and the size of the pots, there may be phenotypic differences between materials grown in a greenhouse and in field conditions.
A9: Thanks for your observation. You are right in the differences that genetic materials can show in greenhouse conditions and in the field, this work is a first approach to these genetic materials from the quinoa germplasm collection because in the field it has been observed that the accessions present mixtures that are reflected in the variations of the qualitative and quantitative characteristics, we will make the evaluation in the near future of the materials under field conditions.
- This.....
A10: Thanks for your observation. In this new version R1, we realized this correction.
- Monthly precipitation of 2 mm? In the greenhouse? Precipitation or irrigation?
A11: Thanks for your observation. We remove this data in the R1 version because it has nothing to do with work.
- Replace yield for grain weight per plant
A12: Thanks for your suggestion.
- Comment. Considering the high coefficient of variation, In the discussion, put that this was a factor that limited the experiment.
A13: Thanks for your comment. Indeed, these high coefficients of variation were due, among other things, to the fact that the genetic materials showed significant differences between them, such as the case of grain weight per plant, which was highly variable. In addition to the variation observed in quantitative characteristics, especially those related to yield.
- Replace yield for grain weight per plant.
A14: Thanks for your suggestion.
- Value (y=62.02g0. This statement gives the impression that 62.02 is the average value.
A15: This is the average weight of the grains per plant (yield) of the individuals of the Quinoa Peruana genetic material, see in table 3.
- Similarly, this statement give us a impression that 2,57 is the average value.
A16: Indeed, it is the average of the seed diameter of the evaluated individuals of the Amarilla de maranganí genetic material (GD = 2.57 mm), see in table 3.
- Adjust formationg in the title of table 3.
A17: Thanks for your observation. The title of the table is according to the instructions for authors of the journal.
- Table 3. How are the statistical differences among genotypes?
A18: Thank you for your observation. In this new R1 version, we adjusted Table 3 with averages, standard deviation, and coefficients of variation for the quantitative variables of the thirty quinoa accessions. Additionally, we show the Tukey test (p <0.05) to better present the statistical differences between accessions. The previous results show the high segregation in the phenotypic characteristics of the evaluated accessions, and they corroborate the results obtained with the other statistical analyzes applied.
- Adjust formating.
A19: Thanks for your observation. In this new version R1, we realized this correction.
- Are in bold. Statistical analysis would be better.
A20: Thanks for your suggestion. Table 3 was adjusted to show the statistical differences between accessions, the minimum and maximum values ​​of the variables between accessions are shown in bold. The other statistical analyzes are shown in the analysis of variance in Table 4 and the multivariate analyzes in Figures 1 and 2.
- Df 2 in the table four means that the experiment had 3 repetitions.
A21: Thanks for your observation. The experimental design used was a randomized complete block design (RCB) with three blocks and nine repetitions or individuals for each material. We in the R1 version adjusted the materials and methods section to clarify the experimental design and the number of plants evaluated. The degrees of freedom Df 2 correspond to the blocks. In the R1 version we adjusted the materials and methods section to explain in more detail the experimental design and the number of plants evaluated.
- Adjust formatting
A22: Thanks for your observation.
- indicate the correlation code in the text in figure 1.
A23: Thanks for your observation. We changed the format of the figure to make it easier to observe the correlations between the variables.
- Figures 2 a and b are too small. Increase the size of these two figures. You can divide in figure 2 and figure 3 or increase size both figures. It is difficult to visualize this figure. Increase the size of it.
A24: Thanks for your observation. In this new version R1, we realized this correction.
- it is difficult to check these data in the figure 2a and 2b.
A25: Thanks for your observation. In this new version R1, we realized this correction.
- Made it instead of male it.
A26: Thanks for your observation. In this new version R1, we realized this correction.
- Morphological instead of morphoagronomic.
A27: Thanks for your observation. In this new version R1, we realized this correction.
- Twenty per cent of... instead of 20%.
A28: Thanks for your observation. In this new version R1, we realized this correction.
- Replace process for stage.
A29: Thanks for your observation. In this new version R1, we realized this correction.
- Ninety five per cent instead of 97%
A30: Thanks for your observation. In this new version R1, we realized this correction.
- 209. Eight percent instead of 80%
A31: Thanks for your observation. In this new version R1, we realized this correction.
- increase size of fig. 5
A32: Thanks for your observation. In this new version R1, we realized this correction.
- Morphological instead of morpho agronomic
A33: Thanks for your observation. In this new version R1, we realized this correction.
- Weight of grains per plant instead of yield.
A34: Thanks for your suggestion.
- Grain weight per plant
A35: Thanks for your suggestion.
- Grey instead of gray, I guess.
A36: Thanks for your observation. In this new version R1, we realized this correction.
- Morphological instead of morpho agronomic.
A37: Thanks for your observation. In this new version R1, we realized this correction.
- grain weight per plant instead of yield.
A38: Thanks for your suggestion.
- Discuss if the temperature, light and humidity conditions inside the greenhouse and the target environment are similar.
A39: Thank for your suggestion. In this new version R1, we realized the adjustments to the discussion.
- Grain weight of grain per plant replacing yield.
A40: Thanks for your suggestion.
- In general, the materials...
A41: Thanks for your observation. In this new version R1, we realized this correction.
- And...
A42: Thanks for your observation. In this new version R1, we realized this correction.
- Replace behavioral for trend.
A43: Thanks for your observation. In this new version R1, we realized this correction.
- Allow instead of allows.
A44: Thanks for your observation. In this new version R1, we realized this correction.
- Replace by morphological
A45: Thanks for your observation. In this new version R1, we realized this correction.
- Replace morpho agaronomicc for morphological.
A46: Thanks for your observation. In this new version R1, we realized this correction.
- Replace morf. agronomic for morphological
A47: Thanks for your observation. In this new version R1, we realized this correction.
- I still prefer MM before results.
A48: Thanks for your observation. The materials and methods is according to the instructions for authors of the journal.
- Measurements inside greenhouse?
A49: Thanks for your observation. The conditions of light, temperature and humidity remained constant, however, there are no data on the behavior of these variables during the entire phenological cycle of the experiment in the greenhouse, we show the climatic conditions of the region where the greenhouse is located.
- According to Analisys of variance (df) (Table 4) you had 3 repetitions. You may have sub samples in the same pot.
A50: Thanks for your observation. The experimental design used was a randomized complete block design (RCB) with three blocks and nine repetitions or individuals for each accession. We in the R1 version adjusted the materials and methods section to clarify the experimental design and the number of plants evaluated. The degrees of freedom D2 correspond to the blocks, that is, in each of them there were three individuals for each accession. In the R1 version we adjusted the materials and methods section to explain in more detail the experimental design and the number of plants evaluated.
- grain weight at what grain moisture?
A51: When the plant reaches physiological maturity, the grain has low humidity at that time, the grain weights were taken.
- Adjust text position on the table.
A52: Thanks for your observation. The title of the table is according to the instructions for authors of the journal.
- Table 1. Adjust coordinates position on the table.
A53: Thanks for your observation. In this new version R1, we realized this correction.
- 387. Twenty one...instead of 21
A54: Thanks for your observation. In this new version R1, we realized this correction.
- L Adjust formatting in title and at the bottom of tables.
A55: Thanks for your observation. The table is according to the instructions for authors of the journal.
- any color scale?
A56: We use the color scale based on the RGB scale.
- better weight of grains per plant and hopefully yield in the field
A55: Thanks for your suggestion.

Reviewer 2 Report
See attached file "Plants_quinoa_MS_review.pdf"

Author Response
Respuesta a los comentarios del revisor 2
“Phenotypic Characterization of Quinoa (Chenopodium quinoa Willd.) for the Selection of Promising Materials for Breeding Programs” by: Elsa Helena Manjarres-Hernández, Diana Marcela Arias Moreno, Ana Cruz Morillo-Coronado, Zaida Zarely Ojeda-Pérez, Agobardo Cárdenas-Chaparro.
- Line 51: throughout the paper, the authors use of the terms/phrases “genotype” and “capable of adapting” (also adapt, adaptable) in a manner that is vague and/or that contradicts their use of “genotype” or vice versa. This makes it is very difficult in review to understand the objectives and the criteria on which accession measurements and observations were judged.
A1: Thank you for your observation. In version R1, we consider the 30 quinoa materials evaluated as accessions, to avoid confusion with the terms, in addition, table 1 shows the origin of these accessions, which have not undergone any normal improvement process and are conserved in germplasm collections or by the farmers themselves. In the department of Boyacá, there is no improvement program aimed at identifying accessions with morphological and yield characteristics that allow responding to the needs of quinoa farmers, producers, and consumers. We adjusted Line 51 “However, information is lacking on the morphological characteristics and grain yield of these accessions”.
- Line 56: here the authors indicate that it’s “essential to efficiently (use) genetic variability to increase the productivity of crops under different environmental conditions.” Applying the term “genotype” to the accessions in this light implies looking at genetic variability present among the accessions and prioritizing those that perform well on some basis, presumably grain yield. However, the study is conducted in a greenhouse/common garden such that there is no environmental variation to against which to look for “genotype”-associated adaptive climatic differences. So, presumably, the authors’ must be looking for accessions (and inappropriately referring to them as genotypes) with promising range(s) of trait variability associated with yield from which to start a program of genetic selection.
A2: Thanks for your observation. The main objective of the study was to determine the phenotypic variation existing in the thirty accessions of quinoa selected under controlled conditions and thus determine which are the most variable qualitative and quantitative morphological characteristics and that contribute more to the discrimination of materials to obtain a selection index that allows me to validate it in other quinoa germplasm under field conditions.
- Lines 349-367: In the Discussion section, especially these paragraphs exhibit either confused thinking by the authors and/or make confusing use of terminology rendering interpretation of the results and understanding of the study aims nearly intractable. Four accessions – Q. primavera, Peruana, real, and Amarilla de marngani – were identified as outstanding materials according to the selection index. I suspect this may be simply because these four accessions had the least variability for traits deemed favorable and associated with high yield whereas the others lacked a similar level of genetic uniformity for yield associated traits (no appropriate variance data for the individual accessions is provided by the authors to say). Yet, in Line 355ff, the authors indicate that the high variability in the descriptors employed in the study are very useful both to increase productivity (grain yield, presumably?) but also to identify capacity to adapt to different environmental conditions. But which is it? Uniform favorable traits or variability in the traits addressed by the descriptors? Is the former true for some (the chosen four?) while the latter is true for the others? For the former, the common garden context of the experiment makes no sense if the authors are looking for environmentally adaptive traits among the accessions. For the latter, here again there is no quantitative evidence presented as to the variability of traits within the accessions – the authors only report %CV of measurements across the entire set of 30 accessions.
A3: Thanks for your observation. Table 3 was adjusted to show the statistical differences between accessions. The selection of the four accessions Quinua amarilla de maranganí, Quinua peruana, Quinua primavera and Quinua real was made based on the proposed selection index where grain yield, plant height, grain diameter, and color. Characteristics such as plant height are important because the low growth of the plants facilitates agronomic work, especially during harvest, more homogeneous seed sizes with higher seed diameters (2.4 to 2.6mm) can facilitate threshing and the use of machinery for post-harvest processes, and the white grain, color are the most desirable characteristics for the commercialization of quinoa in Colombia. However, the software used (RindSel program) generates the weights to the selected variables, allowing the selection of accessions to not be biased by the researchers. Tables 3 and 4 present the descriptive statistics of averages, standard deviation, Tukey’s test (p <0.05), coefficient of variation, and the analysis of variation for the characteristics in whose analysis the phenotypic variation existing in the evaluated accessions can be inferred.
On the other hand, the basis for genetic improvement programs is variability, if it does not exist, no selection can be made since all individuals respond in the same way to the evaluated conditions. Therefore, the phenotypic variability associated with qualitative or quantitative morphological characteristics will allow the selection of accessions that respond to the needs of farmers, producers, and consumers. In addition, from the accessions selected in this study, genotype-by-environment interaction studies can be proposed in the main producing departments in Colombia.
- In sum, the manuscript requires a major revision throughout prior to acceptance in which the authors provide to clarity and precision about what they are aiming to assess and explicit justification for why their experimental approach can provide evidence relevant to that assessment.
A4: We appreciate the constructive and helpful comments of the reviewer in this regard.
The cultivation of quinoa is an excellent alternative to guarantee food security in developing countries. In Colombia, quinoa has been little studied at a morphoagronomic level, in addition, there is no certified seed or registered varieties; given its yield potential and exceptional nutritional characteristics, in Colombia, there are some germplasm collections, which have not been characterized. Also in the field, it can be observed that the materials that are cultivated present mixtures which are evidenced in the segregation of the morphological characteristics and the little uniformity in the grain. That is why the objective of this work was to lay the foundations to start a quinoa improvement program that allows farmers to have certified planting material and with the characteristics that allow increased productivity. Thirty quinoa accessions from the seed collection of the department of Boyacá were selected to study them under controlled conditions to allow them to express this phenotypic variation and thus be able to select the accessions that presented the best morphological and yield characteristics, taking into account that the characters that the farmer is more interested in them. Therefore, a selection index was proposed in which a greater weight was assigned to said characters. Our results indicated that the variables that most contributed to the phenotypic variation were the length of the panicle, the yield, the diameter and color of the seed, and the weight of 1000 seeds. The proposed selection index made it possible to identify Quinua amarilla de maranganí, Quinua peruana, Quinua primavera and Quinua real as those accessions that have the potential to increase productivity but need a more efficient selection process in breeding programs quinoa. We believe that these findings will be important for the region and future work to strengthen the quinoa production chain in Colombia.
- Major methodological concerns: 1) Use of common garden:
- If the aim is to compare accessions (“genotypes”) to determine those with suitable traits associated with yield for the purpose of suiting environmental/climatic variation a common garden lacks any dependent variation – in fact it’s designed to prevent the needed variation to make such an assessment. b. CG is appropriate if the aim is to grow out accessions that lack genetic uniformity confusingly referred to as “genotypes” – this much is implied by stating that farmers in Colombia whence the accessions were collected - grow mixed crops of quinoa and that no prior selection explicitly for genetic improvement has occurred on the accessions – to identify those with mean values for traits correlated with yield – this seems to be more in line with the presented analyses, but again, if so, this needs to be much more clearly and explicitly stated.
A5: You are right regarding the adaptability of these genetic materials because according to the methodological design used, we cannot infer anything about it precisely because the environmental conditions were controlled. Therefore, we adjusted throughout the document that the characterization carried out was limited to only the morphological and not the morphoagronomic.
Furthermore, to avoid confusion for readers, we clarify that we evaluated 30 accessions and not genotypes. The experimental design used in this research was based on works such as those of García et al., 2020 who analyzed the growth and morphophysiological behavior of the quinoa cultivars Blanca Soracá (BS), Blanca Jericó (BT), and Tunkahuan (T) in a region of Colombia. The study was carried out under greenhouse conditions and allowed to recognize that the three cultivars of quinoa presented different times of phenological development, in addition, these cultivars of quinoa expressed productive behaviors that were associated with early and late cycles, showing differences in the yield and weight of seeds.
Another work that we took into account for the experimental design was that of Sosa et al., 2017 who, under greenhouse conditions, evaluated varieties of quinoa to develop a scale of phenological stages for quinoa based on the BBCH system, it should be noted that we with the design We proposed the phenological characterization of the materials although these data are not presented in the manuscript.
- García, M., Stechauner, R., Garcia, J., and Ortiz, D. 2020. Analysis of the growth and morpho-physiological performance of three cultivars of Colombian quinoa grown under a greenhouse. Revista de Ciências Agroveterinárias 19 (1): 73-83. DOI: 10.5965/223811711912020073
- Sosa V, Brito V, Fuentes F, Steinfort U (2017) Phenological growth stages of quinoa (Chenopodium quinoa) based on the BBCH scale. Annals of Applied Biology 171(1): 117–124. https://doi.org/10.1111/aab.12358
2) Sample sizes, variance, and bias:
- It does not seems unreasonable to me that the authors expect 9 replicates with one measurement/observation per replicate, i.e. 9 observations, per accession within the RCB to be adequate to acquire meaningful data. This is especially the case if trying to detect accession (“genotype”) effects among accessions that lack genetic uniformity (within the accession population). A set of ~30 random observations per accession would seem to be a reasonable starting point absent a convincing presentation of or reference to theoretical sample size curve to capture a trait distribution that demonstrates otherwise.
A6: Thanks for your observation. Studies of morphoagronomic characterization of germplasm in quinoa and other cereals have used a small sample size, such as the work of García et al., 2020, where they characterized three cultivars of quinoa under greenhouse conditions with an experimental design where 20 individuals per material were evaluated, which allowed them to recognize the most productive materials and with the most outstanding grain characteristics for commercialization.
In the work of Alanoca and Machaca, an agromorphological characterization of 10 quinoa accessions was carried out in Cochabamba Bolivia, they used an experimental design in randomized complete blocks (RCB) with 4 repetitions of each treatment or accession/variety, and samples were taken in five plants to morphological characterization, and with this, it was shown that the accessions were grouped into two groups, the earliest materials and those with the highest grain yield.
In cereals such as barley Morillo et al., 2020 carried out an agromorphological characterization of 83 accessions where they used a completely randomized block design (RCB), and the measurements were made on 10 individuals per accession, through this methodology it was possible to identify accessions with characteristics desirable agronomic. Therefore, we consider that the sample size and the experimental design used are valid to obtain reliable inferences.
- García, M., Stechauner, R., Garcia, J., and Ortiz, D. 2020. Analysis of the growth and morpho-physiological performance of three cultivars of Colombian quinoa grown under a greenhouse. Revista de Ciências Agroveterinárias 19 (1): 73-83. DOI: 10.5965/223811711912020073
- Alanoca, C., and Machaca, E. 2015. Caracterización agromorfológica de 10 accesiones y variedades de quinua (Chenopodium quinoa ) en condiciones del Valle Alto de Cochabamba. Revista científica de investigación INFO-INIAF 1(5): 21-29.
- Morillo, A., Pulido, W., and Laiton, Y. 2020. Caracterización agromorfológica de cebada (Hordeum vulgare) en el Municipio de Chivatá Boyacá, Colombia. Biotecnología en el Sector Agropecuario y Agroindustrial 18(2): 103-116.
- Again, with high ambiguity as to the actual study aims, it is dismaying that the authors present no accession-wise measures of sample variance (SD or SE, the latter probably appropriate for the number of observations/sample). Without knowing the sample variance, the individual sample means (and ranges) themselves are of entirely unknown representative value for making meaningful comparisons. Again, this is especially true of samples from genetically nonuniform populations. If, on the other hand, an objective is to identify populations with high trait variation as having potential for adaptability (i.e. standing genetically associated trait variation from which to perform selective breeding), the sample variances are just as important to know in conjunction with the need for larger, adequate sample sizes.
A7: Thank for your suggestion. In this new R1 version, we adjusted Table 3 with averages, standard deviation, and coefficients of variation for the quantitative variables of the thirty quinoa accessions. Additionally, we show the Tukey test (p <0.05) to better present the statistical differences between accessions.
- %CV across the RCB study data reported by the authors seem both inadequate and inappropriate for their (hard to know) study objectives. This is a variance metric generally applied to assess sample measurement (often instrumental) uniformity across a common set of common samples. In this case, the samples themselves are of anything but nominally common materials – this is why the sample variance is what needs to be represented and interpreted in the study.
A:8 Thank for your suggestion. Studies of morphoagronomic characterizations of quinoa using different sources of germplasm from both wild and cultivated materials analyze the CV values ​​since it is a measure of variation that allows giving an idea of ​​the behavior of the variables under the evaluated conditions.
In the work of Maliro and Njala, 2019 they used the CV to evaluate the behavior of quinoa varieties in different environments of Malawi. The genotypes were arranged in a completely randomized block design with four repetitions and present between genotypes the coefficient of variation of the quantitative variables. Bhargava et al., 2007 use CV in evaluating the morphological and qualitative characteristics of 27 Chenopodium quinoa germplasm lines and 2 lines of C. berlandieri subsp. nuttalliae which was carried out in north Indian subtropical conditions, selection criteria were established to increase yield and harvest index and correlations between yield and yield components. On the other hand, Alanoca and Machaca, 2015, also used the CV to establish the genetic variation and the adaptation of quinoa genotypes in the Upper Valley of Cochabamba, some reported data revealed that the Variation coefficients (CV) oscillate around 4% (days to harvest) to 54% (yield).
Anyway, according to your suggestion, we adjusted Table 3 with the standard deviation and Tukey test (p <0.05), which corroborate the results obtained with the other statistics.
- Maliro, M., and Njala, A. 2019. Agronomic performance and strategies of promoting Quinoa (Chenopodium quinoa Willd) in Malawi. Ciencia e Investigación Agraria 46(2): 82-99, DOI 10.7764/rcia.v46i2.2143
- Alanoca, C., and Machaca, E. 2015. Caracterización agromorfológica de 10 accesiones y variedades de quinua (Chenopodium quinoa) en condiciones del Valle Alto de Cochabamba. Revista científica de investigación INFO-INIAF 1(5): 21-29.
- Bhargava, A., Shukla, S., and Ohri, D. 2007. Genetic variability and interrelationship among various morphological and quality traits in quinoa (Chenopodium quinoa). Field Crops Research 101: 104–116, DOI:10.1016/j.fcr.2006.10.001
- It is unclear (unstated) whether the authors sowed on one seed/replicate/accession in the RCB or whether multiple seeds were sown and but one plant was chosen for measurement.
A9: Thanks for your observation. In the R1 version, we adjust in the materials and methods section to give clarity in the selection of the plants that we evaluate as follows: “The germination of the seeds was carried out in the nursery with a mixture of humus and peat in a 2:1 ratio. By material 16 alveoli were sown where three seeds were placed that were taken randomly, after 20 days of growth when the seedlings had six true leaves, they were transplanted to the greenhouse beds, thinning was carried out when more than two plants grew per alveolus. The accessions were sown under a randomized complete block design (RCB) of three plants per block (three blocks) for a total of nine repetitions for each accession, with conventional agronomic management”.
- If the former, we need to know how the individual seeds themselves were sampled from the accession seed pools, especially since seed traits (weight, diameter, color, shape) are subjects of analysis. Hopefully there was some randomization method and not arbitrary choice which would be inherently subject to confounding bias. If randomly chosen, the problem with small sample size (i.e. N = 9) remains concerning if there is a broad distribution of measurable/observable values for any, some, or all of the traits of interest.
A10: Thanks for your observation. Before taking the accessions to the greenhouse, the seeds were characterized (weight, diameter, color, and shape) and germination tests were carried out (Data not shown). After evaluating the viability of the embryo, we made our seedbed by taking the seeds randomly. Therefore, we did not select the seed. We consider that N = 9 is valid to obtain reliable inferences.
- Si se sembraron múltiples semillas de accesiones en cada uno de los 3 bloques replicados, ¿cuántas? Es de esperar que no en masa, ya que presumiblemente una alta variación en el rasgo de la semilla de la accesión haría que KWT sea posiblemente incognoscible y / o de valor limitado para proporcionar un factor de ajuste de tamaño de muestra adecuado. Finalmente, si se sembraron varias semillas en cada réplica, ¿cómo se seleccionó la planta de cada réplica para la medición? Una vez más, existe un potencial excesivamente alto de que el sesgo del investigador sea un factor aquí.
A11: Gracias por tu observación. Como se mencionó anteriormente, las nueve plántulas para las mediciones fueron seleccionadas al azar de las 16 plántulas obtenidas por accesión cuando fueron trasplantadas al invernadero, precisamente para evitar sesgos en la selección de plantas dentro de las accesiones.

Round 2
Reviewer 2 Report
Thank you for your consideration and thoughtful response to the feedback I provided in the first review.
The revised manuscript is much improved in communicating with clarity the nature of the subjects of study (accessions vs. "genotypes"), the study objectives (finding selectable variation in traits associated with greater yield potential), and the study design/methods (common garden to observe within accession variation under controlled environmental conditions). As a reviewer, my background is in plant population genetics, genetic mapping & quantitative genetics working only tangentially with evaluation of cultivars/varieties within crop species, therefore I especially I thank you for your response and references supporting the study design, sample sizes, and statistical metrics in similar studies. I hope that the reading from outside the direct field of study was helpful in improving communication to a broader audience. In that regard, thank you for inclusion of variance (S.D.) metrics and Tukey multiple comparison (MC) analysis of trait measurements within accessions. Having requested these, I do suggest that the authors reconsider the presentation of these results and analysis in Table 3 as it is quite unwieldy and busy in its current form. I suggest eliminating the repetitious S.D. columns by collapsing each trait into a single column with "mean (SD)" in each cell (e.g. upper right cell would be "128.6 (14.35)" for Quinoa real, PH) with the reporting convention explained in the table legend. This may be enough to improve the presentation, but the MC results might better be shown in a separate table/matrix from the means and variance results table. Lastly, I'm not sure that presenting a global ANOVA (Table 4) adds anything meaningful given that there were one or more pairwise accession mean differences for all nine quantitative traits shown in the MC. If in a similar vein to reporting global CV for each trait, it is meant to show support for the study design in identifying differences, it would help to state as much explicitly, though again, I think the MC results show this to be true.
Overall, the revised MS is much improved in clarity and meaning of results for a general reader and I recommend publication with consideration of the Table 3 and 4 suggestions.
Author Response
Response to Reviewer 2 Comments
Point 1: Thank you for your consideration and thoughtful response to the feedback I provided in the first review.
The revised manuscript is much improved in communicating with clarity the nature of the subjects of study (accessions vs. "genotypes"), the study objectives (finding selectable variation in traits associated with greater yield potential), and the study design/methods (common garden to observe within accession variation under controlled environmental conditions). As a reviewer, my background is in plant population genetics, genetic mapping & quantitative genetics working only tangentially with evaluation of cultivars/varieties within crop species, therefore I especially I thank you for your response and references supporting the study design, sample sizes, and statistical metrics in similar studies. I hope that the reading from outside the direct field of study was helpful in improving communication to a broader audience. In that regard, thank you for inclusion of variance (S.D.) metrics and Tukey multiple comparison (MC) analysis of trait measurements within accessions. Having requested these, I do suggest that the authors reconsider the presentation of these results and analysis in Table 3 as it is quite unwieldy and busy in its current form. I suggest eliminating the repetitious S.D. columns by collapsing each trait into a single column with "mean (SD)" in each cell (e.g. upper right cell would be "128.6 (14.35)" for Quinoa real, PH) with the reporting convention explained in the table legend. This may be enough to improve the presentation, but the MC results might better be shown in a separate table/matrix from the means and variance results table. Lastly, I'm not sure that presenting a global ANOVA (Table 4) adds anything meaningful given that there were one or more pairwise accession mean differences for all nine quantitative traits shown in the MC. If in a similar vein to reporting global CV for each trait, it is meant to show support for the study design in identifying differences, it would help to state as much explicitly, though again, I think the MC results show this to be true.
Overall, the revised MS is much improved in clarity and meaning of results for a general reader and I recommend publication with consideration of the Table 3 and 4 suggestions.

Response 1: We greatly appreciate each of your observations, which allowed us to improve the writing. In version R2 we adjust table 3 according to your suggestion. And we remove table 4 that corresponds to the ANOVA, we agree that it does not provide given that there was one or more pairwise accession mean differences for all nine quantitative traits shown in the Tukey multiple comparison MC.
In addition, we removed the global report from the CV and adjusted the results section according to Tukey multiple comparisons (MC). We did not generate a new table for the MC test because it would be very similar to table 3.
